# Autonomous Visual Navigation for a Flower Pollination Drone

**Dries Hulens** [1,†], **Wiebe Van Ranst** [1,*,†] , **Ying Cao** [2] **and Toon Goedemé** [1,*]

1.  EAVISE (Embedded and Artificially intelligent VISion Engineering), KU Leuven, 2860 Sint-Katelijne-Waver, Belgium; dries.hulens@kuleuven.be
2.  Magics Technologies NV, 2440 Geel, Belgium; ying.cao@magics.tech
*   Correspondence: wiebe.vanranst@kuleuven.be (W.V.R.); toon.goedeme@kuleuven.be (T.G.)
†   These authors contributed equally to this work.

**Abstract:** In this paper, we present the development of a visual navigation capability for a small drone enabling it to autonomously approach flowers. This is a very important step towards the development of a fully autonomous flower pollinating nanodrone. The drone we developed is totally autonomous and relies for its navigation on a small on-board color camera, complemented with one simple ToF distance sensor, to detect and approach the flower. The proposed solution uses a DJI Tello drone carrying a Maix Bit processing board capable of running all deep-learning-based image processing and navigation algorithms on-board. We developed a two-stage visual servoing algorithm that first uses a highly optimized object detection CNN to localize the flowers and fly towards it. The second phase, approaching the flower, is implemented by a direct visual steering CNN. This enables the drone to detect any flower in the neighborhood, steer the drone towards the flower and make the drone's pollinating rod touch the flower. We trained all deep learning models based on an artificial dataset with a mix of images of real flowers, artificial (synthetic) flowers and virtually rendered flowers. Our experiments demonstrate that the approach is technically feasible. The drone is able to detect, approach and touch the flowers totally autonomously. Our 10 cm sized prototype is trained on sunflowers, but the methodology presented in this paper can be retrained for any flower type.

**Keywords:** pollination drone; visual servoing; two-stage approach; neural network

## 1. Introduction

World food consumption is projected to increase in the coming decades [1]. Together with a declining insect population [2], and given the roles bees play in the pollination of crops [3], this is a big concern for world food security. To have a successful crop, a farmer might rely on the surrounding ecosystem, artificially introduce a bee colony to pollinate their crops or rely on some other biological pollination method. However, due to collapse of surrounding ecosystems, regulatory difficulties such as protection from invasive species, and the circumstances of an artificial factory environment, interest in human made pollination methods has grown. Currently, pollination robots are already being used [4,5].

These methods use mobile robot arms together with some kind of pollination brush to go from flower to flower, distributing pollen from plant to plant. Using such a robot arm however does have some limitations: (i) the way a farm is laid out might for instance not easily allow a robot to pass between different crops, (ii) the type of produce might not lend itself to be touched by a robot arm, or (iii) the field might be too big or steep to build infrastructure for such a mobile ground robot. Presently, some crops are even planted above each other in vertical farms, which makes it impossible for a wheeled robot to pollinate [6]. All of this means that there is a lack and a certain need for a more generic and versatile pollination method.

In this paper, we present work towards a drone equipped with a camera and on-board processing to autonomously pollinate flowers. The approach we present enables a drone to autonomously detect flowers and approach them using a two-stage deep learning approach.

As a preliminary use case, we focus here on sunflowers. This is a deliberate choice, as our final goal would be to develop a nanodrone (with a size of a few cm), that can pollinate small crop flowers, such as strawberry flowers, cucumber flowers, etc. As the hardware at such small scale is not obtainable yet, in this work we demonstrate the technology at a slightly larger scale, where the large size of sunflowers matches with the scale of the 10 cm mini-drone used.

The three main novelties of this paper are:

1. We demonstrated the deployment of deep learning based computer vision in real-time on-board a resource restricted embedded processing platform.
2. We developed a methodology to train the necessary neural networks with a partially real, partially synthesized dataset.
3. We successfully demonstrated a two-stage flower approaching visual servoing procedure.

In the remainder of this paper, we investigate how to create such a visual navigation system for a pollination drone. In Section 2, we lay out the related work on robot pollination methods, autonomous drone technology and the computer vision techniques we use to steer the drone. In Section 3, we go in depth into the inner workings of our drone navigation approach. In Section 4, we explain the experiments we did to evaluate the effectiveness of our approach and discuss our promising results. We conclude in Section 5. This paper is an extended version of the conference paper [7] we previously presented at ARCI 2022.

## 2. Related Work

In this section we will first present the basics of drone navigation and visual servoing. Then we investigate existing artificial pollination solutions and then go into hardware platforms that can be used for on-board computer vision based navigation.

### 2.1. Drone Navigation

In this section, we will briefly discus the Degrees Of Freedom (DOF) of our drone and its main components. In general, a drone has four or more propellers that deliver thrust to keep the drone into the air. When all propellers deliver an equal amount of thrust and generate an upward force equal to gravity, the drone is hovering. This means that the drone maintains its position in the air. To fly in a certain direction, some of the propellers are slowed down causing a rotation and a translation in that direction. When all propellers spin in the same direction, the drone would start rotating around its own Z-axis. To counter this torque effect, two of the four propellers turn clockwise and the other two turn counter-clockwise as in seen in Figure 1 (left).

The drone that we are using for this paper is a DJI Tello Robomaster TT. This type of drone has six DOFs as seen in Figure 1 (right). When the front and back pairs of rotors (1&2 vs. 3&4) are steered to turn with a different speed, the downward thrust vector turns around the Y-axis. This results in a horizontal force component, causing the drone to translate forward or backwards. This movement is called the *pitch*. A rotation around the X-axis (slowing down the left or right pair of motors) yields a translation to the left or right, called the *roll*. The roll and pitch are directly coupled with a translation left, right, forward and backward. When a diagonal motor pair gets another speed w.r.t. the other pair, a net rotation momentum causes the drone to rotate around its vertical axis. This is called the *yaw*. The Z-axis can be controlled separately by giving all four motors the same speed increase or decrease, to adjust the altitude of the drone.

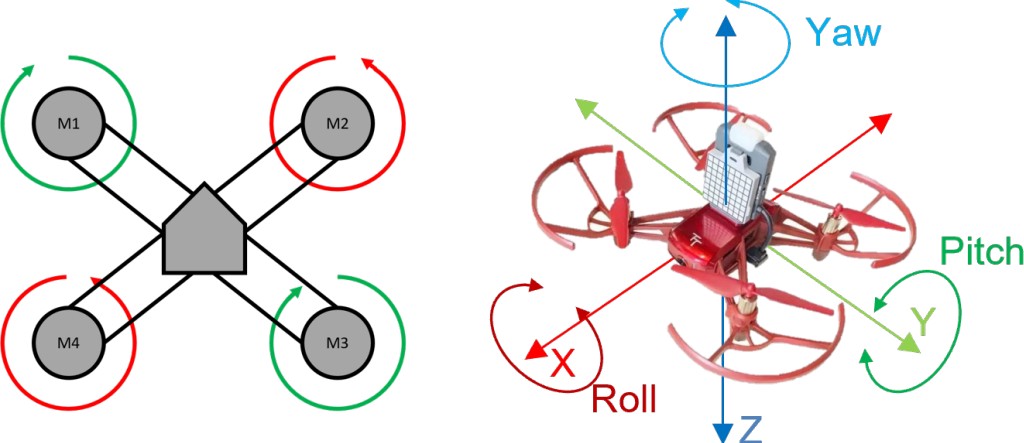

**Figure 1.** (**Left**): A quadrocopter with the directions of the turning propellers displayed. (**Right**): The six DOFs of our drone.

As seen in Figure 2 the heart of a drone is its flight controller. This controller ensures that the drone remains in an idle horizontal position when no control commands are sent to the flight controller, called *hovering*. When control commands are received, the flight controller calculates a proper speed for each of the (in our case four) motors, e.g., when a "go forward" command is received, the flight controller lowers the speed of the two motors in the front which yields a forward movement. Without this flight controller, the pilot should control the four motors independently to make this movement, which is almost impossible. To stabilize the drone, the flight controller tries to maintain its angle in the X, Y and Z plane. To do this, the current orientation of the drone is calculated using two main sensors as explained below.

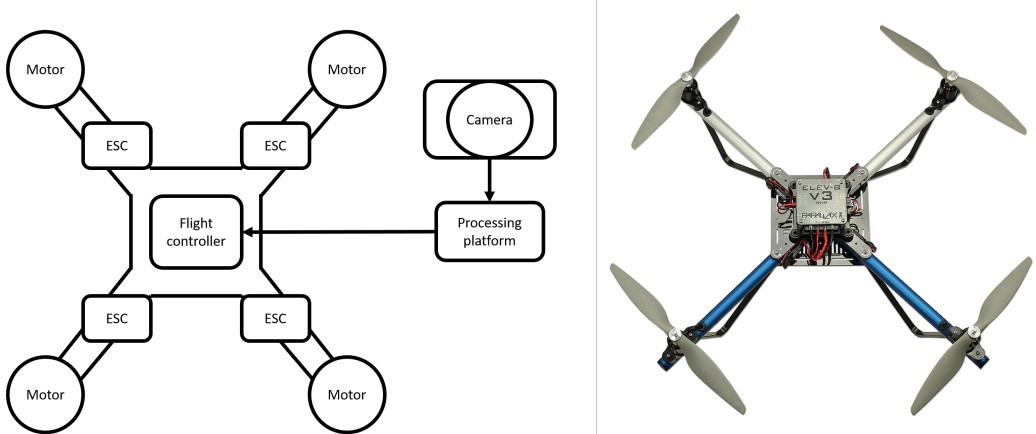

**Figure 2.** The main components of a drone: Flight controller, ESC (Electronic Speed Controller) and motors. We added an extra camera and processing platform to our drone.

The first is the accelerometer, which outputs the acceleration in three directions (for X, Y and Z). This acceleration is expressed in gravity units (1 g = 9.8 m/s$^2$). Thus, when the acceleration sensor is not moved, the output will be 1g for the z-axis and 0g for the X- and Y- axis of the sensor. These values are used to calculate the angle of the sensor in three directions but cannot be used directly to measure the orientation of the drone. The downside of an acceleration sensor is that the output is not reliable on its own. The accelerometer measures inertial force, such a force can be caused by gravitation (and ideally only by gravitation), but it might also be caused by acceleration (movement) of the device. Thus, when the drone is moving, the output of the accelerometer cannot directly be used to

capture the gravity vector. Furthermore, even if the accelerometer is in a relatively stable state, it is still very sensitive to vibration and mechanical noise in general. To deal with these problems, a gyroscope is used to smooth out these errors. A gyroscope measures the rate of changes of the angles in *deg*/s which is less sensitive to linear mechanical movement. Although, the disadvantage of a gyroscope is that it suffers from drift (not coming back to zero when rotation stops). Nevertheless, by fusing the data of both sensors, an accurate angle of the drone can be obtained to help the flight controller stabilize the drone.

Our flight controller also contains a barometer to maintain altitude and a small down-looking camera to maintain a fixed position. The accuracy of these sensors is of less importance in our experiments, since we use Image-based Visual Servoing which does not use global coordinates as further explained. When the flight controller has calculated a velocity for each motor, the velocity is passed to a Electronic Speed Controller (ESC) which drives the motors. As shown in Figure 2 we also equipped our drone with a camera to capture images which are used by the algorithm running on the embedded processing platform. Our task is to develop software that interprets the images and generates control commands for the flight controller to steer the drone. This software should run in real-time on an embedded platform, mounted on-board.

### 2.2. Visual Servoing

Visual servoing is the technique we use to steer our drone, using information from the camera. This technique can also be used to control e.g., a robot arm, where the camera can be mounted on the end effector of the robot arm (eye-in-hand), or somewhere else fixed on the frame (eye-to-hand). In the case of a drone, the camera is fixed to the drone itself, and thus an eye-in-hand system. There are two main visual servoing techniques: Position-based Visual Servoing (PBVS) and Image-based Visual Servoing (IBVS).

In PBVS, the 3D position of the object (in our case a flower, in the case of a robot arm an object to grasp) is calculated w.r.t. the camera. Then, a control command is calculated to go from the current 3D position of the end effector to the goal position. When the object is moving, its new 3D position and the 3D position of the end effector should be estimated in every frame, to recalculate a new path to move. The disadvantage of this approach is that the accuracy of the system is sensitive to calibration errors.

In IBVS, only the 2D position of the object is estimated in the frame and used to calculate an error between the current 2D position and the desired 2D position for every frame. This error can then be used to control the actuator, or drone. We use IBVS in this paper, since our camera is mounted on the drone itself and we want to position our drone w.r.t. the flower. The 3D position in the world is of less importance. Indeed, our main goal is to approach the flower, even if it moves slightly. A more detailed description of PVBS and IBVS can be found in [8].

### 2.3. PID Control Loops

In this subsection, we illustrate the control loop we used to steer the drone smoothly with an example. The drone can be steered in different ways to approach a flower. In this example, we will only use the pitch to position the drone at a fixed distance, e.g., 4 m, w.r.t. a flower. When the distance between the drone and flower increases, the pitch should be controlled to fly forward and vice versa. When controlling the pitch, the drone will rotate around its Y-axis (see Figure 1) with a certain angle resulting in a net horizontal force and hence a translation of the drone. The flight controller enables us to simply send the desired velocity of the translation, which is then automatically recalculated to a certain angle for the pitch.

A possibility is to control the pitch binary such that the pitch velocity is maximal, e.g., 10 m/s when the distance is larger than 4 m, and −10 m/s when the distance is smaller than 4 m. Indeed, this will result in a very jerky movement of the drone. A better way to control the pitch is proportional to the error between the desired distance and the current distance as in Figure 3. Here, the measured distance is 4.2 m, resulting in an error of 0.2 m.

This error is multiplied by a proportional ($K_p$) gain (e.g., $5\,\text{s}^{-1}$) resulting in a pitch velocity of 1 m/s. Using only the $K_p$ gain, the control loop is formulated as in this equation:

$$v = K_p \cdot E(t) \tag{1}$$

where $v$ is the velocity and $E(t)$ the error.

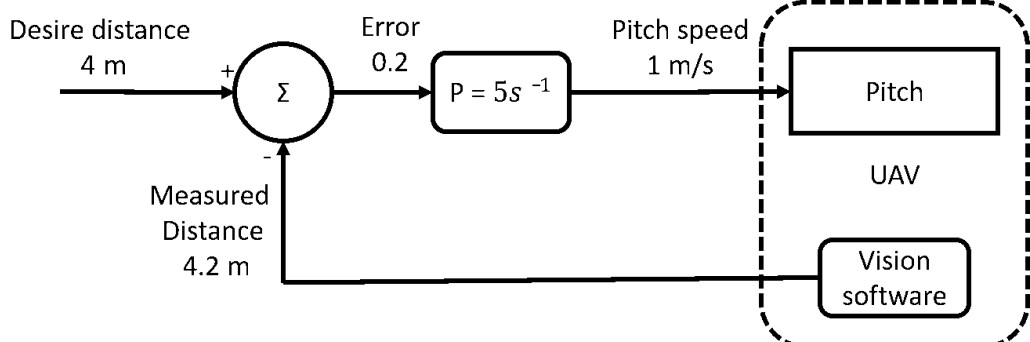

**Figure 3.** A proportional control loop. The error between a desired and measured value is calculated and multiplied with a scale factor to generate a control value.

By increasing or decreasing the $K_p$, the desired distance can be reached faster or slower. So, why not making $K_p$ as high as possible, e.g., $100\,\text{s}^{-1}$? When $K_p$ is high, the velocity of the pitch will also be high, even at a small error. An error of e.g., 0.1 m will result in a velocity of 10 m/s and the drone cannot be stopped in time (when reaching 4 m) due to its momentum of inertia, and will overshoot and oscillate around its desired distance. A high proportional gain can even lead to an unstable system, where the output of the P controller increases every frame until the maximum speed of the drone is reached. We indeed better limit the maximum velocity of the drone to a safe value. Making $K_p$ smaller can reduce the oscillation, but it will take longer to decrease the error and it is even possible that the drone cannot reach its desired distance. When P is small, e.g., $0.1\,\text{s}^{-1}$ at 4.2 m, the error will be 0.2 m and the velocity 0.02 m/s. Such a small velocity will not move the drone enough and the desired distance cannot be reached.

To overcome this problem, the integral term (I) is introduced as in Figure 4. The I term integrates the error over time. When the error is too small to move the drone, this error will remain and the I term sums this error over time. This means that the longer the error exists, the higher the output of the I term will be, resulting in an increase in velocity, which eventually will move the drone until the error becomes 0. The control loop can now be formulated as:

$$v = K_p \cdot E(t) + K_i \int_0^t E(t) \cdot dt \tag{2}$$

The velocity $v$ is the sum of the P and I terms.

As mentioned before, we want our $K_p$ gain as high as possible to react as fast as possible. A high $K_p$ gain induces overshoot. To damp this overshoot a derivative (D) term is introduced as in Figure 5. This D term will look at the rate of change of the error over time. When the drone comes closer to its goal distance of 4 m, the error will become smaller very fast. The D term senses this fast change and starts to generate an inverse control value, to slow down the drone even faster (braking), resulting in a minimal overshoot. The final PID control loop is formulated as:

$$v = K_p \cdot E(t) + K_i \int_0^t E(t)dt + K_d \cdot \frac{dE(t)}{dt} \tag{3}$$

The final velocity $v$ is the sum of the P, I and D terms.

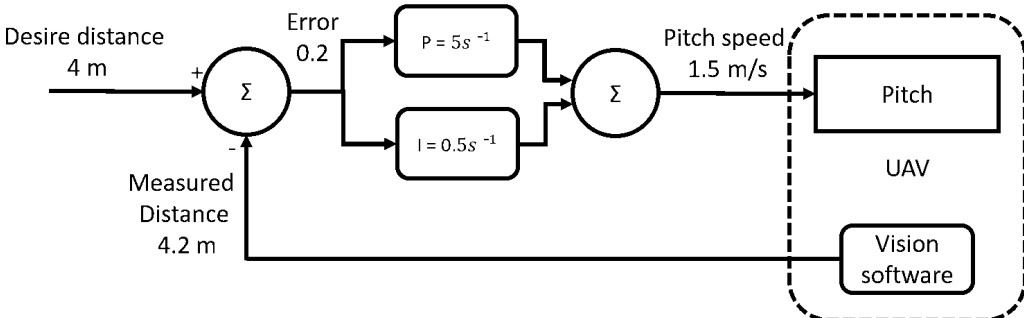

**Figure 4.** A proportional-integral control loop. The integral factor ensures that the error can decrease to 0.

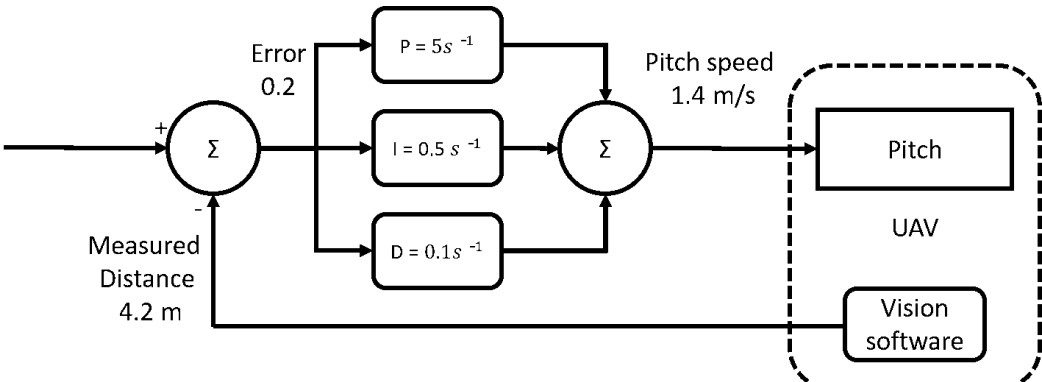

**Figure 5.** A proportional-Integral-Derivative control loop. The D factor damps the overshoot induced by P and I.

For each of the axes, such a PID loop is implemented to control the drone smoothly. Of course, $K_p$, $K_i$ and $K_d$ are tuned for each axis as explained in Section 3.4.1.

A PID control loop is a closed-loop controller, since the action from the controller is dependent on the process output. This can also be combined with an open-loop controller (feed-forward controller). Here, knowledge of the inertia and acceleration of the drone is fed forward in the control loop to make the output even more stable. We did not use this combination, since there is no inertia model of our drone publicly available. However, in the future, this model can be approximated doing real-life experiments.

PID control loops are not the only control loops that can be used to steer a drone. In [9], Fuzzy Logic was used to control a drone to fly waypoints. The advantage of using Fuzzy Logic is that the reaction of the controller on a certain input can be formulated more precisely for different states of the system. Although, since different states of the system can be tuned separately, a large number of parameters should be fine-tuned compared to a PID loop. Hence, for most systems, a PID control loop is preferred above Fuzzy Logic.

Another approach is to use iTaSC (instantaneous task specification and control) [10] as a control loop. iTaSC is a systematic constraint-based approach to specify complex tasks of general sensor-based robot systems, such as a drone. The advantage of using iTaSC is that it can automatically derive controller and estimator equations from a geometric task model, that is obtained using a systematic task modeling procedure. We did not use iTaSC in this paper, due to the more time-consuming integration w.r.t. PID control loops. Nevertheless, it would be interesting to use this approach in the future. Indeed, iTaSC can control several axes of the drone automatically to accomplish a specific movement using a geometric model of the drone, where in our approach all axes should be controlled independently by different control loops.

### 2.4. Related Work on Artificial Pollination

As mentioned earlier, world food consumption is projected to increase significantly in the coming decades [1]. To address this challenge, more efficient ways of producing food should be developed to feed this growing demand. A possible solution using technological means is to aid nature with the pollination of crops and plants. In recent research, robots were already developed to automatically pollinate crops and flowers by using a robotic arm on a base equipped with wheels [5,11]. For the detection of the flower, they use Inception-V3 [12] together with color segmentation. This is a common approach for detecting flowers as seen in [13] where they combine a CNN and SVM to predict an even more accurate segmentation of the flowers. The downside of a robotic arm on a wheeled platform is its size and maneuverability.

In [14] a drone was used to pollinate the flowers. This drone has multiple advantages over a wheeled robot, but was way too big (50 cm × 50 cm) to do the extremely precise job of pollinating flowers. In this work we developed a small drone (10 cm × 10 cm) with on-board processing power which can estimate the position and angle of the flower. Furthermore, this drone can autonomously fly towards the flower in order to precisely pollinate the flower.

### 2.5. Hardware Platforms for On-Board Computer Vision Based Navigation

We investigated many hardware platforms that would fit our criteria for on-board computation of deep learning based computer vision tasks. To meet our requirements, the platform should be: (i) light-weight, (ii) low-power, (iii) easy to deploy and integrate with an existing drone, and (iv) able to run state-of-the-art computer vision algorithms with a sufficiently high frame rate.

For the most part, this rules out higher power mini-computer devices such as the NVIDIA Jetson series. The STM32F746 or other ARM Cortex M7 based platforms provide a deep learning library (CMSIS-NN) to run deep learning applications on the CPU architecture. The Ambiq Apollo3 Blue development board is able to run Tensorflow Lite models mostly targeted at voice and gesture recognition, but not computer vision. The Greenwave GAP8 is a RISC-V processor optimized to run deep learning models in a multi-threaded way. The Kendryte K210 processor is also based on a RISC-V core, but in addition to that also contains a deep learning accelerator that is able to run YOLO [**?** ] models at up to 20 fps. Due to its performance and relative light weight, we decided that the Kendryte processor best fits our use case.

From a machine learning point of view, we need models that are able to do two types of tasks: detection and classification (direct steering). These areas are a very active field of research, and many pipelines are available, each with their own trade-offs. Discussing all current techniques would lead us too far from the scope of this paper. Instead, we will discuss the techniques that are available on the chosen platform here. The Kendryte processor has support for the Mobilenet [16] family of backbones. This is a backbone especially optimized for low-power mobile devices, and is able to save a tremendous amount of computing power by using depthwise convolutions. Since the publication of the Mobilenetv1 paper, many additions to the network have been proposed as well [17]. However, due to the platform we choose, we are limited to Mobilenetv1. The classification head uses a simple cross entropy loss. In the detection stage we use a modified detection network, together with a Mobilenet to do fast inference. The detector makes bounding box predictions with only a single pass over the network, as opposed to region-based detectors like Faster-RCNNN [18] and Masked-RCNN [19]. Due to their nature, generally speaking, single stage detectors are faster than region-based ones. The authors in [20] did a complete survey on the current state of object detection algorithms.

## 3. Autonomous Drone Navigation

In this section we go further into detail about the inner workings of the visual navigation system for our pollination drone. We create an artificial sunflower dataset containing

images of real, as well as artificial and virtual sunflowers. We use the low power Kendryte K210 hardware platform which is able to run quantized Mobilenet based networks mounted on a commercially available DJI Tello Talent drone. A two stage flower approaching technique is developed to successfully touch the flower. In the first stage, we detect the flower from a distance between 8 and 0.8 m using a convolutional detection network. In the second stage (0.8–0 m) we use an end-to-end network to predict the center of the flower in the camera feed. To guide the drone towards the center of the flower, a PID control loop is used to convert the location of the flower to steering commands for the drone. In this section, we explain all of these components one by one.

### 3.1. Flower Dataset

To train the computer vision neural networks, we created an artificial dataset based on images of real sunflowers, artificial (synthetic) sunflowers and virtually rendered sunflowers. An important aspect we take into account is that for the drone to successfully pollinate a sunflower, it should approach the sunflower in a line perpendicular to the sunflower. In order to be able to do this, we need to estimate the angle of gaze orientation of the sunflower to enable us to correct for this when we are planning the drone's trajectory. This means that in each image the angle of the flower w.r.t. the camera should also be annotated.

For the virtual flowers, we simply render them at different angles, both in horizontal and vertical direction. For the real and synthetic sunflowers, we made a recorder device that is able to rotate the flower automatically over these two degrees of freedom while taking pictures of the flower. Both are illustrated in Figure 6. The generated images are then pasted on images from the Places dataset, more specific on images from Japanese gardens [21] to use as background. A total of 5660 annotated images with multiple flowers, and 4500 zoomed-in images on one flower were generated this way. Examples of these images can be seen in Figure 7.

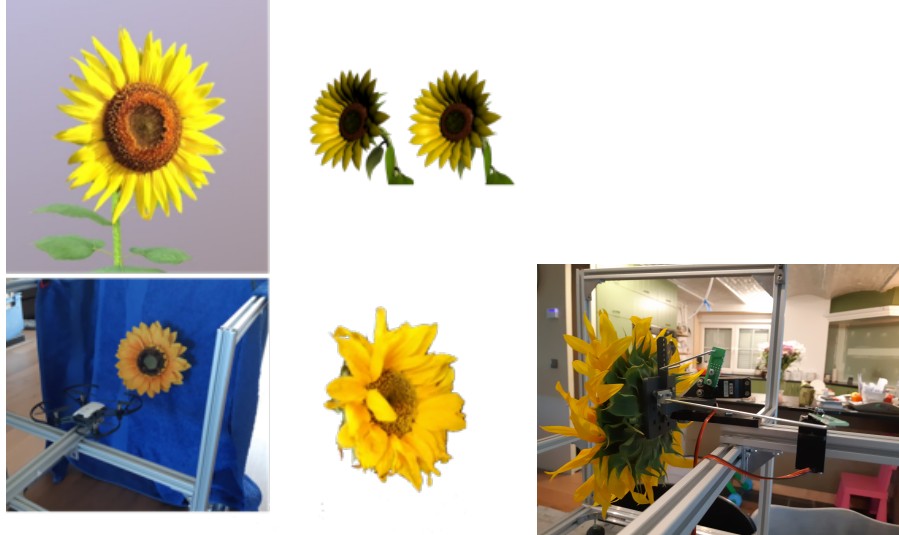

**Figure 6.** Acquisition of the training data. Top: 3D models of flowers are rendered at different angles and positions. Bottom: our chroma key set-up that is used to acquire images from both artificial (**left**) and real (**center**) sunflowers, with the drone's camera, using a 2 DoF rotator device (**right**).

### 3.2. Hardware Platform

Since the precise steering the drone has to perform, processing of the video feed should happen on-board such that the delay between receiving images and correcting the drone's position is minimal. This implies the need for a small and light-weight processing board capable of performing real-time image processing. We choose to use a Maix Bit development-board which contains a Kendryte K210 (RISC-V) processor. This board

measures 53 mm × 25 mm and weighs 25 g. Furthermore, this board is equipped with a camera, resulting in a minimal image receiving delay.

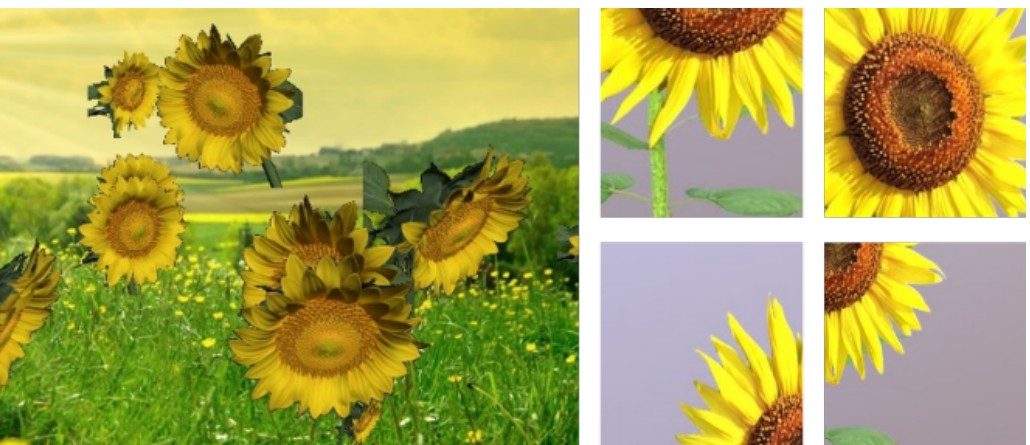

**Figure 7.** (**Left**): 3D generated sunflowers for the flower detector. (**Right**): Generated images for direct visual servoing approach.

The camera feed is directly read by the Maix Bit on which we either run the end-to-end network or the detection network. The Maix Bit communicates with the Tello drone via its on-board ESP32 microcontroller. In addition to the camera, we also have a ToF depth sensor (VL53L1X) to measure the distance from the flower in the final approach. A detailed overview of our setup is shown in Figure 8. Figure 9 shows the drone carrying the Maix Bit.

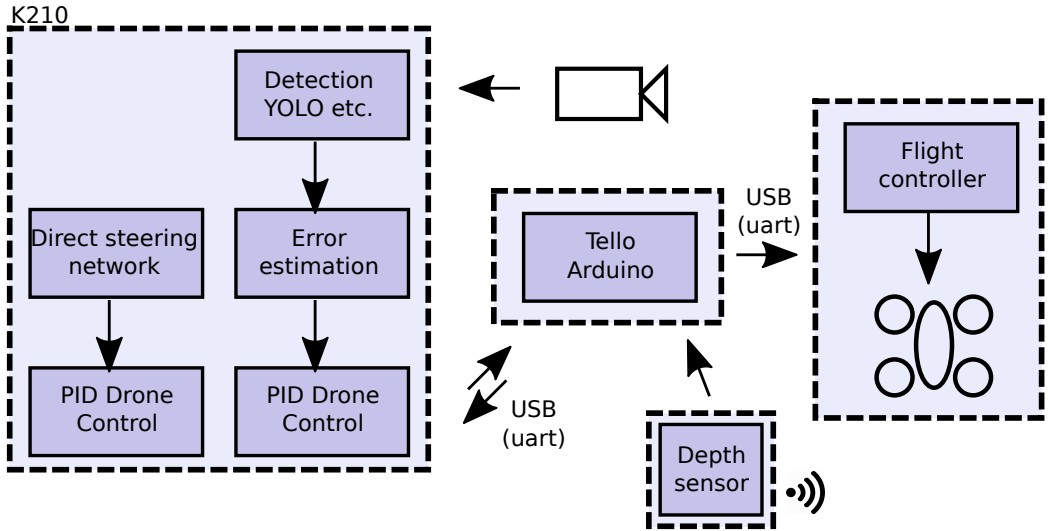

**Figure 8.** Overview of our hardware setup.

### 3.3. Hybrid End-to-End and Detection Approach

Figure 10 gives an overview of the piece-wise visual servoing approach we developed to steer the drone towards a flower. As explained before, the vision part of our approach plan is divided into two stages. Figure 10 gives an overview of our visual servoing approach. In the first stage we run a CNN for flower pose estimation: position, size (indication of the distance) and orientation. This model steers the drone to fly towards a position close to the flower (approximately 80 cm), directly facing it. This stage is done in three steps: (i) a forward movement until the drone is at the right distance from the flower, (ii) an altitude change such that the drone is at the flower's height and (iii) an encirclement motion around the flower, until the drone faces the flower directly. In the second stage, we used an image-based visual servoing network in an end-to-end approach such that it directly

outputs steering commands towards the sunflower's position. This model is used for the final approach, the pollination touchdown.

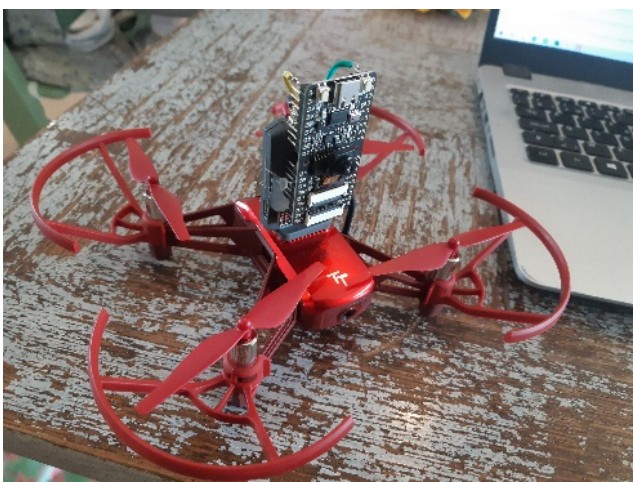

**Figure 9.** Our drone prototype carrying the Maix Bit.

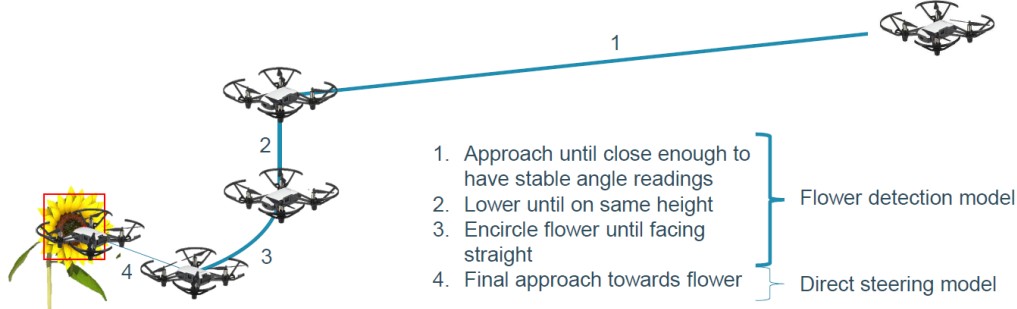

**Figure 10.** Piece-wise visual servoing approach overview.

### 3.3.1. Detection Stage

In the first stage (distance of 8 m to 0.8 m) we use a MobileNet [16] detector trained on our own flower dataset. The position of the flower is used to steer the drone such that the flower is always in the center of the drone's view. In this stage, the size of the detected flower is used as a distance measurement. Figure 11 illustrates the measured properties of the flower that our detector outputs, and the steering dimensions associated with them.

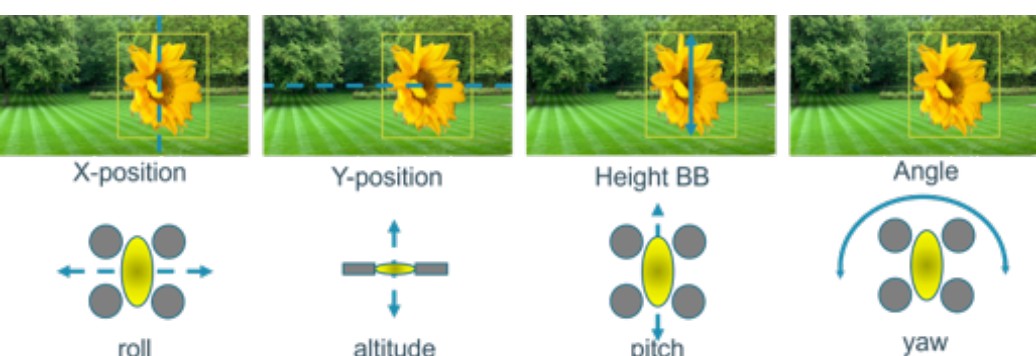

**Figure 11.** The four properties that our detection model perceives of the flower and their related steering dimension.

As indicated, we altered the CNN architecture to additionally output the horizontal angle of the flower, illustrated in Figure 12. This predicted angle is used to steer the drone

towards a 0° angle w.r.t. the flower. When the drone approaches the flower and the size of the flower exceeds 60% of the frame-height, the flower becomes too big to be reliably detectable, and we switch to the second visual servoing stage.

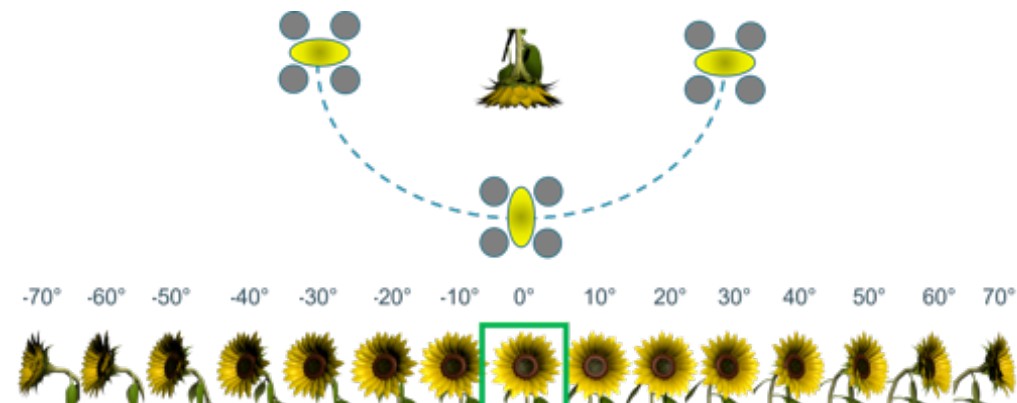

**Figure 12.** Illustration on the horizontal angle detection of the flower.

### 3.3.2. Direct Visual Servoing

When the drone approached the flower close enough for the final descent, our detection model of the previous phase does not work anymore. That is because the flower's size exceeds the field-of-view of the camera. In this stage we use an end-to-end network based again on a Mobilenetv1 architecture trained for classification that directly outputs steering commands (up, down, left, right or center). This network is trained on zoomed-in images of a flower (see Figure 13). During the final approach to the flower, we use a VL53L1X ToF distance sensor to measure the distance between the drone and the flower with high accuracy. The pollination rod in front of the flower measures 8 cm. When the distance becomes smaller than 8 cm we assume the rod touched the flower and the pollination took place.

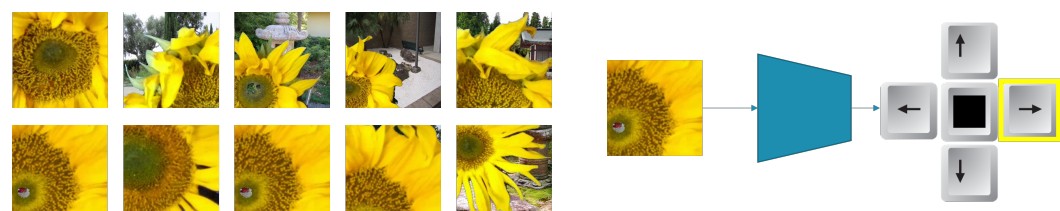

**Figure 13.** (**Left**): example images from the direct visual servoing image dataset. (**Right**): The proposed direct visual servoing approach.

After pollinating the first flower, we plan to let the drone do a random maneuver (consisting of a backward and upward movement, random horizontal rotation and an adjustable horizontal movement). Then, the drone will be in a new random position from where it can start again approaching the closest sunflower. This random motion will eventually make it possible to visit multiple flowers in a field.

### 3.3.3. Optimization of the Neural Networks

In order to run neural networks on a resource-restricted embedded device, optimization is necessary. From a memory point-of-view, the biggest challenge is storing all model weights locally. This is essential to guarantee real-time execution. However, also the activation map memory storage and the amount of compute operations needs to be restricted, even for an already very efficient and lightweight model as the MobileNet architecture we use for both models. To cope with this, we opted to post-quantize the parameters of our TFLite model from 32 bit floats to 8 bit ints. Additionally, the NNCase compiler included within the MaixPy Kendryte toolchain optimized our models further using operator fusion

and memory optimization. This enabled us to squeeze both MobileNetV1 models together into the 8 MB RAM of the tiny Kendryte K210 hardware platform we use.

### 3.4. PID Control Loop

To transfer the position of the flower into steering commands for the drone, four different PID loops are used, one for each of the drone's axes. The goal of the PID loops is to center the flower in the drone's view and steer the drone such that the angle between the drone and the flower reaches zero degrees (ideal for pollination). To position the drone such that the flower is in the center of the screen, the Y-coordinate of the flower is used to move the drone up or down (Altitude) while the X-coordinate is used to steer the drone to the left or right (Roll). The angle of the flower is used to control the drone's rotation around its Z-axis (Yaw). When the Yaw is changed, consequently, the flower will also move to the left or right in the image, which will be again compensated by controlling the Roll. To approach the flower, the size of the detection is used to move the drone forwards or backwards (Pitch). When the distance becomes smaller than 0.8 meter, we switch to an end-to-end stage where the time of flight distance sensor is used to control the Pitch instead.

#### 3.4.1. Tuning of the PID Loops

Each axis of our drone is steered by a control loop that gets its information from the vision software. In this section we explain how we tuned $K_p$, $K_i$ and $K_d$ for each control loop. A first tuning of the PID parameters happened in simulation. DJI provides a simulator to learn how to fly with their drones and the (virtual) drone in the simulator reacts very similar to the real one, since DJI made a dynamic model of the Tello. Since we cannot insert flowers and a camera in the simulation environment, we replaced the flower with a specific GPS position and developed software to fly to that GPS position and stop when it arrives. First $K_i$ and $K_d$ were set to 0 and $K_p$ was chosen so that the drone was just not oscillating around the goal GPS position. Then $K_i$ was increased to reach the GPS position (make error 0), with the result of a small overshoot. Next $K_d$ was increased to damp this overshoot. Of course, since our software induces small delays while measuring the depth, the position of the flower and the angle of the flower, the PID values are also fine-tuned in real-life. For this we developed an android-app to tune the PID values in real-time. Nevertheless, this fine-tuning took a lot of time since each time a specific loop changed, its PID settings had to be adjusted. To further overcome oscillations around the setpoint we inserted a dead-zone in our system. This is a zone around the setpoint where the drone should not correct its position. A threshold value for the corrections was also chosen empirically. These PID control loops run at a fixed speed of 50 Hz, all in separated threads, which is fast enough to provide the flight controller with data.

## 4. Experiments and Results

In this section, we will summarize the experiments we have conducted with the developed visual navigation system.

### 4.1. Flower Detection

The foremost task of the on-board vision processing is detecting the flower in the image, such that the drone can steer itself towards it. For this, we implemented the MobileNet flower detector. Figure 14 shows some qualitative detection results, both when multiple flowers are in view, and at various distances. When multiple flowers are in view of the drone, it selects the largest and most confident flower detection to first navigate towards.

Table 1 shows a comparison of different models we trained for the combined task of flower detection and angle estimation. For there models, we started from pretrained weights on ImageNet, and trained for 50 k steps. The alpha factor controls the width of the network. This is known as the width multiplier in the MobileNet paper. If alpha < 1.0, it proportionally decreases the number of filters in each layer. The detection result is evaluated using mAP, the mean average precision. For each of the detection thresholds, we evaluate

how many detections overlap enough (using an intersection over union threshold of 0.5) with ground truth bounding boxes. These precision results are averaged over the detection thresholds, yielding the AP (average precision) score for each class. After optimization of the models (see Section 3.3.3), we chose the MobileNetV1 model with alpha factor 0.5, because that is the best performing model that fits in the hardware.

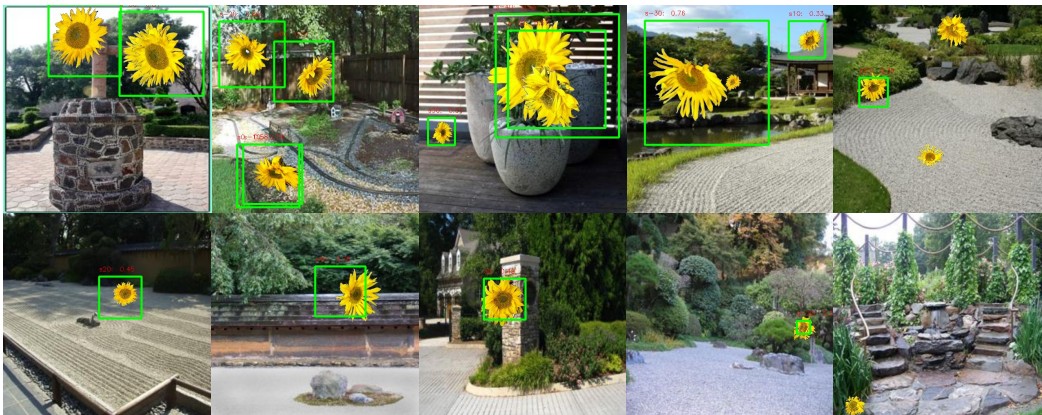

**Figure 14.** Illustration of detection results both when there are multiple flowers in view (**upper**), and when there is only one flower in view (**lower**).

**Table 1.** Experimental results of different models for flower detection and angle estimation.

| Model | Alpha Factor | Resolution | Detection AP |
|---|---|---|---|
| SSD + MobileNetV1 | 1.0 | $320 \times 240$ | 0.68 |
| SSD + MobileNetV1 | 0.5 | $320 \times 240$ | 0.61 |
| SSD + MobileNetV1 | 0.25 | $320 \times 240$ | 0.51 |
| SSD + MobileNetV2 | 1.0 | $320 \times 240$ | 0.77 |
| SSD + MobileNetV2 | 0.5 | $320 \times 240$ | 0.66 |
| SSD + MobileNetV2 | 0.35 | $320 \times 240$ | 0.62 |

On the real-world evaluation dataset, we evaluated the average precision of the angle detection results of our chosen model. As explained above, our model uses $10°$ angular bin classes for angle detection. The final mAP is computed as the mean AP over all these classes. We reached a mAP of 0.36 on this test dataset (note that the different angle classes in our dataset make our dataset more challenging; when doing real-world tests this accuracy proved sufficient).

In Figure 15 we evaluate the detection accuracy against the distance the flower is moved from the camera (equivalent to the size of the flower in the image), we can see that flowers further away from the camera are harder to detect. Moreover, if flowers are too close by, the detection also begins to fail. That is the main reason for the second model (direct steering) model we propose, where no explicit object bounding boxes are computed.

### 4.2. Direct Steering Model

In the second phase, when the drone is close enough to the flower, the navigation is taken over by our second computer vision model. As described before, this model directly outputs motion commands, based on the centeredness of the (visible part of the) flower in the camera image. First, we trained a MobileNetV2 architecture on our direct steering training dataset (see Figure 13). Unquantized, using Float32 number representations, this model reaches 86.2% accuracy, after quantization to Int8 70.0%. A MobileNetV1-model yields 70.5% in Float32 and 69.7% in Int8. On the *difficult* test part of our direct steering dataset, this MobileNetV1 model achieved 53% accuracy.

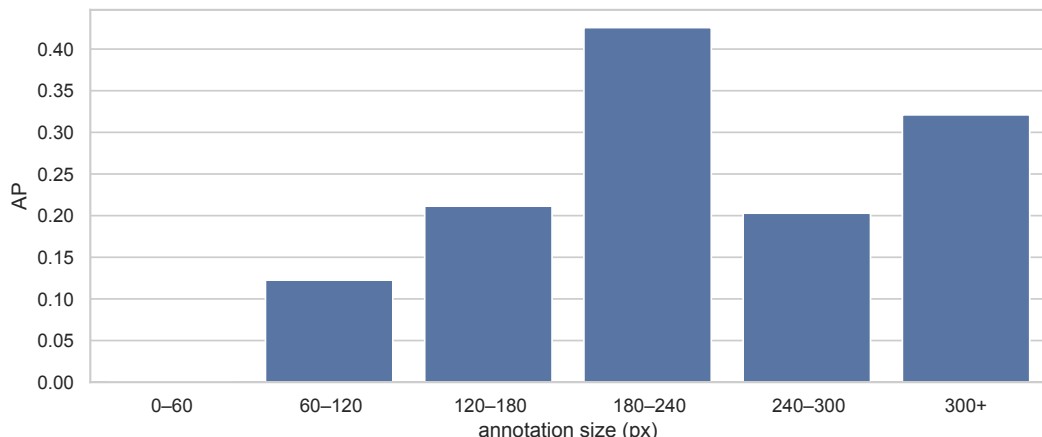

**Figure 15.** Influence of the distance from the sunflower (smaller flowers are assumed to be further away from the camera) on the detection accuracy.

In order to test this model separately, we wrote a control loop that steers the drone such that it keeps facing the flower directly while maintaining a fixed distance of 20 cm. This intensively uses our on-board running direct steering model, and additionally the time-of-flight sensor for the distance measurement. A demonstration of this experiment can be seen at https://youtu.be/oxx7KPLV804 Accessed on (9 May 2022). It is clear that the accuracy of the direct steering model is sufficient to steer the drone accordingly.

*4.3. Visual Navigation*

In order to validate the full system, we conducted an experiment evaluating the success-rate of steering a drone towards the pollinating position for a sunflower. We repeated this experiment 24 times where the drone took off from a distance of 5 m to 8 m, flew to the flower, touched it and flew back. In total, we reached a success-rate of 87%. In the other 13% of non-success, a false detection was the cause of failure. For the successful navigation trials, the average time to reach the pollination position is 31.5 s, with a standard deviation of 15.3 s.

The maximum frame rate of which images could be processed on the Maix Bit is approximately 30 fps for the direct steering model and 20 fps for the detection network. We also tested the minimum frame rate needed to smoothly steer the drone towards the flower since a slower frame rate results in less power consumption. The minimum framerate needed to smoothly approach a flower is measured at 12 fps.

Table 2 provides URLs to videos of six of these trial flights. Under different light conditions, our drone was able to successfully navigate towards the flower. Sequence 3 also demonstrates that our system can cope with outdoor conditions. However, because of the small weight of the drone, we notice that too much wind is certainly a challenge for precise navigation.

**Table 2.** Demonstrations of the full navigation approach.

| Sequence nb | Conditions | Remarks | Video URL (Accessed 9 May 2022) |
|---|---|---|---|
| 1 | indoor | cluttered environment | https://youtu.be/quX5HhVyR3g |
| 2 | indoor | demo at Dubai World Expo | https://youtu.be/u13j3sPgDlE |
| 3 | outdoor | low wind conditions | https://youtu.be/ixOCjHggUw4 |
| 4 | indoor | difficult light conditions | https://youtu.be/DZh7zHVQJqM |
| 5 | indoor | initial heading away from flower | https://youtu.be/Lq7TR70cJJk |
| 6 | indoor | long search for flower | https://youtu.be/AhhI29ofmr0 |

In earlier experiments (e.g., sequences 1 and 2 in Table 2), we noted that the final approach towards the flower is functional, but that it centers the camera with the center of the flower, and not the pollination rod we mounted on it. In later experiments (e.g., sequences 3–6), we corrected this vertical bias, such that the pollination rod is now exactly steered towards the flower's center. Figure 16 shows four of these successful trials. As seen in the bottom row, the drone's pollinating rod exactly touches the center of the flower now.

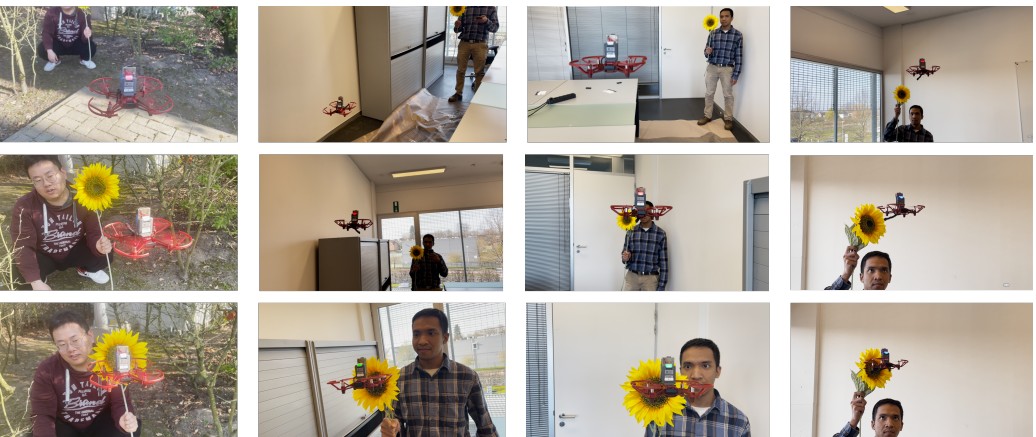

**Figure 16.** (Columnwise) Successful navigation trials towards the pollination position.

### 4.4. Discussion

In this paper, we focused on steering an autonomous drone towards a flower for pollination. We know that in reality, a successful pollination occurs only when a bee lands on the head of the sunflower and drags pollen from its legs onto the stigma in the head of the plant. Just touching part of the flower will not induce pollination. Therefore, in order to extend the current prototype towards a real pollination device, it needs a pollination tool that has the same effect of this biological action, such as a brush, vibrating rod or even a pressured air gun. With our simple pollination rod, visible in the demonstration videos, successful sunflower pollinations are impossible. However, the developed navigation system is necessary in any case, including for other pollination implements.

## 5. Conclusions

In this paper we developed the navigation system for a pollination drone that detects flowers, estimates the angle and flies towards the flower in a two stage approach. Different experiments were performed to evaluate all the parts of this approach, yielding promising results. We managed to run all processing on-board, resulting in a fully autonomous drone with a minimal delay between taking images and the control of the different degrees of freedom. Real world test demonstrate a navigation success rate of 87%.

Furthermore, in this work we demonstrate that, using currently available hardware, it is possible to autonomously fly a drone towards a flower and maneuver it in a way similar to what a bee would do to pollinate a flower. Although the present prototype is trained for sunflowers, the methodology can easily be retrained for any other flower species based on our hybrid training approach with simulated and real images.

However, it remains to be seen how well methods like these would scale up to replace actual bee populations, and what other environmental impacts a dwindling insect population can have on the survival of humanity, and the surrounding ecosystems it relies on. A discussion should be had on how much we want to rely on technological solutions for our own food security, and what role humanity should play in interfering in existing ecosystems.

**Author Contributions:** Conceptualization, D.H., W.V.R., Y.C., T.G.; methodology, D.H., W.V.R., Y.C., T.G.; software, D.H. and W.V.R.; validation, D.H. and W.V.R.; investigation, D.H., W.V.R., Y.C. and T.G.; resources, Y.C., T.G.; data curation, D.H. and W.V.R.; writing—original draft preparation, D.H., W.V.R. and T.G.; writing—review and editing, D.H., W.V.R. and T.G.; visualization, D.H., W.V.R. and T.G.; supervision, Y.C., T.G.; project administration, Y.C., T.G.; funding acquisition, Y.C., T.G. All authors have read and agreed to the published version of the manuscript.

**Funding:** This work has been supported by the company MAGICS, the FWO SBO project OmniDrone under agreement S003817N and the Flemish Government under the AI Research Program.

**Data Availability Statement:** Due to the commercial nature of this research we are not able to make the sunflower dataset publicly available. However the methods described in this paper could be used to create a dataset with similar properties.

**Conflicts of Interest:** The authors declare no conflict of interest.

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
