# Peer review of "Autonomous Visual Navigation for a Flower Pollination Drone"

_machines, doi:10.3390/machines10050364_

Round 1

Reviewer 1 Report

This article presents a small drone that is able to autonomously pollinate flowers. Two-stage visual servoing algorithm is applied to detect flowers and fly towards it. The article is good in its application purpose. Thank you for your efforts in developing this work and the results shared in your article. I have a few comments that I would like to share if you please:

  1. In line 309 of the article, it is written that the range of the first stage is 8m-0.8m, and the sunflowers in datasets shown in the article (Fig. 6 and Fig. 7) are very significant. If the UAV is far away from the flower (for example, 7-8m), can it really detect the sunflower correctly? In other words, does the dataset contain multi-scale images of different sizes?
  2. The scene shown in Figure 7 contains multiple sunflowers to be pollinated. What is the route strategy of UAV pollination at this time?
  3. In the abstract, line 314 and other parts of the article, it is mentioned that you have optimized and altered the architecture of CNN, but the article does not explain what changes have been made to Mobilenet v1.
  4. The article explains PID loops in detail, but does not explain in detail how to set the dead-zone in line 360.
  5. The author carefully attached a video link to show the work in detail. In the first video link in line 379 of the article, it is noted that there is a certain height difference between the camera and the pollination rod, resulting in that the center of the field of vision is not the center of the pollination rod. If the flowers are smaller, this error will have a great impact. How to compensate the deviation caused by flowers of different sizes?
  6. Could you please add some experiments on the robustness analysis of PID and the deviation caused by parameter selection
  7. Could you please supplement the experimental results of target detection in the test dataset, and the experimental results in complex scenes (for example, the same scene contains many flowers).

Thank you very much.

Author Response

We thank the reviewer for the detailed study of our paper and the constructive comments given. In the limited time available, we adapted the paper accordingly. These changes certainly increased the quality of the paper a lot, a fact for which we are very thankful to the reviewer.

  1. In line 309 of the article, it is written that the range of the first stage is 8m-0.8m, and the sunflowers in datasets shown in the article (Fig. 6 and Fig. 7) are very significant. If the UAV is far away from the flower (for example, 7-8m), can it really detect the sunflower correctly? In other words, does the dataset contain multi-scale images of different sizes?

This question is indeed very important. We both added visual as well as quantitative results investigating this dependence of the successful detection of a flower w.r.t. the distance. We added figure 14, illustrating the detection of sunflowers in typical images of our test dataset, giving a visual impression of the capability of our system to detect flowers far away. Moreover, we conducted an analysis on the detection results on our test dataset, yielding the bar graph in figure 15. As can be seen, smaller (and thus farther away) flowers have less chance to get detected. Important to see is that figure 14 shows very few false detections. That means that our drone may sometimes misses to detect a sunflower, but is never flying to another object. This behavior is observed in the (also newly added) demonstration sequence nb 6: Initially, the flower is not detected, causing the drone to rotate around its position until a stable flower detection is present and the actual approach is started.

  1. The scene shown in Figure 7 contains multiple sunflowers to be pollinated. What is the route strategy of UAV pollination at this time?

Indeed, this aspect was not clarified in the text. We added this detail in our text (section 4.1):  When multiple flowers are in view of the drone, it selects the largest and most confident flower detection to first navigate towards. 

  1. In the abstract, line 314 and other parts of the article, it is mentioned that you have optimized and altered the architecture of CNN, but the article does not explain what changes have been made to Mobilenet v1.

Thank you for pointing out that missing item. We added section 3.3.3 to the manuscript, covering the details of the model optimization steps performed.

  1. The article explains PID loops in detail, but does not explain in detail how to set the dead-zone in line 360.

Indeed. The dead-zone is implemented as a simple threshold on the position correction value. If the value is under a predefined threshold, we do not alter the position of the drone. The setting of this threshold value is done empirically, using our smartphone app. We added some information about this detail in section 3.4.1.

  1. The author carefully attached a video link to show the work in detail. In the first video link in line 379 of the article, it is noted that there is a certain height difference between the camera and the pollination rod, resulting in that the center of the field of vision is not the center of the pollination rod. If the flowers are smaller, this error will have a great impact. How to compensate the deviation caused by flowers of different sizes?

We thank the reviewer for pointing that out. This effect was caused by the fact that our direct steering visual navigation model tries to align the flower’s center with the camera. As the camera is mounted high on our prototype, this caused a vertical position error for the pollination rod. For the major revision of this paper, we solved this error by implementing a vertical bias. In the newly conducted experiments (sequences 3 to 6 in table 2, also visible in figure 16), this position difference is taken into account, resulting in more correct “pollinations”, where the rod exactly touches the center of the flower.

  1. Could you please add some experiments on the robustness analysis of PID and the deviation caused by parameter selection

We agree that the robustness of the PID tuning is very important in an application like ours. Unfortunately, in the limited time available for the revision of this paper (10 calendar days, including the Easter weekend), it was not feasible for us to conduct totally new experiments on this. For such experiments, we require an external position measurement system, to track the movements of the drone with different PID parameter settings. This is not available in our lab.

  1. Could you please supplement the experimental results of target detection in the test dataset, and the experimental results in complex scenes (for example, the same scene contains many flowers).

Thank you for this suggestion. We hope that the newly added experimental results (i.c. figure 14 and the demonstration videos mentioned in table 2 and illustrated in figure 16) suffice for this.

Reviewer 2 Report

General Comments:

  1. The paper presents the interesting development of a drone system to pollinate sunflowers.
  2. The development of the system and protocols is well done.
  3. The experimental design has major flaws for a peer-reviewed journal paper that needs to be addressed before this manuscript can be considered for publication.
    1. More details are needed on experimental design. With a well-conceived experimental design, the results will be more meaningful for the reader. Furthermore, a more thoughtful discussion and conclusion can be drawn from the system that was developed herein.
    2. Experimental design should factor in different variables so the readers know the developmental stage that this system is at.
    3. Look at other recent articles for this journal to see how experimental case studies are conducted (e.g., https://www.mdpi.com/2075-1702/10/4/273/htm)
    4. Some items to include:
      1. What were the climatic conditions during the field tests? Do you anticipate climatic conditions to adversely impact results?
      2. How does the wind from the drone during hovering impact the success of pollination?
  • How do the CNNs perform when looking at different types of vegetation and not just sunflowers?
  1. Consider different arrangements of sunflowers (simple grid patterns) to simulate sunflowers planted in rows vs grown in clusters in the field.
  2. How does sunflower orientation (sunflowers are level, angled upward or downward) impact the success rate of pollination?
  3. What statistics can you provide related to timing and the various control measures needed for each successful pollination? (i.e., is the drone struggling and taking a long time to pollinate or is it happening relatively quickly?)
  • If you have multiple objects and other types of vegetation lined up with the sunflowers, how well does the drone determine the sunflowers vs the other objects? Use multiple trials and statistics to share quantitative results.
  1. Attention needs to be paid to grammar and structure throughout.
    1. There are multiple instances of poor paragraph structure throughout. A paragraph should have more than 2 sentences: main idea in first sentence, 2-3 support detail sentences, and 1 transition sentence to next paragraph. Need to fix throughout
    2. Watch run-on and unnecessarily complex sentences throughout.
    3. Also, watch punctuation throughout. There are numerous missing commas after introductory clauses that make the sentences more difficult to read than necessary.

Details:

Abstract:

  1. Line 11: Results from revised experimental case study need to be included in abstract

Introduction:

  1. Line 14: Replace “in” with “is”
  2. Line 14-16: This is an example of a run-on sentence that needs to be fixed throughout.
  3. Line 24-26: For complex lists, consider using (a), (b), (c) to break up the major clauses.
  4. Line 27: Remove jargon (e.g., nowadays) throughout
  5. Line 31: This is an example of comment 4c. Make sure to include commas after transitory clauses throughout (e.g., in this paper)
  6. Line 31-32: This is an example of comment 4a. Make sure to remove these 1-2 sentence paragraphs throughout.
  7. Line 40-43: Use consistent verb tense and construction for all elements of your list (i.e., start with past tense verb)
  8. Line 46: Another example of comment 4c.
  9. Line 50-51: Another example of comment 4a.

Related Work:

  1. Line 55: On-board should be hyphenated throughout
  2. Line 62: Another example of comment 4c.
  3. Line 92: Nice figures throughout. The graphics are easy to follow and avoid unnecessary detail.
  4. Line 116-118: Example of comment 4b. The sentence is unclear as written with the current punctuation. Check for similar issues throughout.
  5. Line 119-120: Position-based and Image-based should be hyphenated throughout.
  6. Line 121 & Line 127: More examples of comment 4c.
  7. Line 133: Another example of comment 4a.
  8. Line 136-138: This should be two separate sentences
  9. Line 215: Use “too” not “to”
  10. Line 221: Deep Learning does not need to be capitalized mid-sentence
  11. Line 221-227: Be consistent with numbered list formatting. consider "To meet our requirements, the platform should be: 1) light weight, 2) low-power, 3) easy ....”
  12. Line 262: Region-based should be hyphenated
  13. Line 263-264: Another example of comment 4a.

Autonomous drone pollination:

  1. Line 255: Make sure to use consistent capitalization across all section headings (e.g., Related Work)
  2. Line 316-318: What happens when the wind from the drone blows the flower or when winds move the flower? How do your algorithms handle this volatility? This needs to be addressed in the manuscript.
  3. Line 360-363: More examples of comment 4a.

Experiments and results:

  1. Line 364-379: See comment 3.
  2. Line 365-366: Need experimental results comparing sunflower detection to other types of flowers/vegetation/objects to determine how well the algorithm has learned sunflowers from other objects.
  3. Line 366: Please explain mAP of 0.36 for the reader in the manuscript.
  4. Line 370: Use “where” not “were”
  5. Line 378-379: The videos show the drone approaching the sunflower but the head of the sunflower is not touched in either case. Is this considered a successful pollination? In reality, a successful pollination occurs when a bee lands on the head of the sunflower and drags pollen from its legs onto the stigma in the head of the plant. Just touching the petals (ray florets) will not induce pollination. This needs to be clarified throughout. Here is a sunflower diagram (https://depositphotos.com/202041350/stock-illustration-parts-sunflower-plant-morphology-flowering.html) and a decent video on the process (https://www.youtube.com/watch?v=klEQG76OG0w). There are obviously more scholarly works that you should reference in this manuscript.
  6. Line 378-379: Also, could your training dataset be improved so that your detection algorithms focus on the sunflower head instead of the sunflower in general? This will more accurately replicate successful pollination.

Conclusion

  1. Line 382-386: This conclusion is misleading based on the brevity and incompleteness of Section 4. I anticipate the conclusions and discussion will be much improved once satisfactory experimental design and results are implemented.
  2. Line 392-394: Well done tying the system development back to the broader context of the problem.

Author Response

We thank the reviewer for the detailed study of our paper and the constructive comments given. In the limited time available, we adapted the paper accordingly. These changes certainly increased the quality of the paper a lot, a fact for which we are very thankful to the reviewer.

First of all, we thank the reviewer for the very detailed writing suggestions. As non-native English speakers, it is very worthwhile for us to have this kind of feedback. We did our best to thoroughly go through the paper and correct these mistakes. It certainly lifted our paper to a higher linguistic level.

The main suggestions and remarks content-wise were the poor experimental validation of the system as a successful pollination device. We totally agree with the reviewer that, when re-reading the original manuscript, the impression was raised that the presented system was able to successfully pollinate a sunflower. But, the latter is not the case. Our contribution lies in the development of a prototype camera-based visual navigation approach for such a pollinating drone, where all image processing is done on-board. We did not study or try to solve issues concerning the actual biological pollination, or even the deployment of such a system on a real-world agricultural situation. We thoroughly changed the manuscript to avoid this misleading interpretation: we changed the paper’s title, and rewrote parts of the abstract, introduction and conclusion to reflect the fact that the navigation system is the part studied in this work.

  • More details are needed on experimental design. With a well-conceived experimental design, the results will be more meaningful for the reader. Furthermore, a more thoughtful discussion and conclusion can be drawn from the system that was developed herein. Experimental design should factor in different variables so the readers know the developmental stage that this system is at. Look at other recent articles for this journal to see how experimental case studies are conducted (e.g., https://www.mdpi.com/2075-1702/10/4/273/htm).

We totally agree that further experimental validation would strengthen the paper. Unfortunately, in the limited time available for the revision of this paper (10 calendar days, including the Easter weekend), it was not feasible for us to conduct totally new experiments. However, we gathered as much as we can and extended the experimental results section substantially:

  • We both added visual as well as quantitative results investigating the experimental variable of the dependence of the successful detection of a flower w.r.t. the distance. We added figure 14, illustrating the detection of sunflowers in typical images of our test dataset, giving a visual impression of the capability of our system to detect flowers far away. Moreover, we conducted an analysis on the detection results on our test dataset, yielding the bar graph in figure 15.
  • We added comparative results of different neural network models (different versions of the MobileNet architecture and settings of the depth multiplication factor alpha), and summarized these in table 1.
  • We added a new subsection discussing experiments with the separate direct steering model (sec. 4.2). We both provide quantitative results on the classification accuracy of the model on a test dataset, as well as provide qualitative evaluation in the form of a video containing a specific experiment with this steering model only.
  • We extended section 4.3 with more details and newly added demonstration experiment videos. We now report on success rate, navigation time, frame processing speed, and demonstrate the system in six different conditions, including difficult lighting and outdoor conditions.

  • Some items to include:
    • What were the climatic conditions during the field tests? Do you anticipate climatic conditions to adversely impact results?
    • How does the wind from the drone during hovering impact the success of pollination?

Thank you for these questions. We understand the answers are interesting to know, but our prototype is in a too early state to do real-life field tests. Most tests were executed in indoor lab environments. The few test that were performed outside, showed that wind is indeed an important factor. The stability of such a lightweight miniature drone is very low in windy conditions. We made all of this clear in the text, as described above. 

  • How do the CNNs perform when looking at different types of vegetation and not just sunflowers?

Thank you for this question. As we have trained the neural networks on sunflowers only, they only will be functional on the trained flower type. As seen in figures 7 and 14,  the background images are gathered from natural landscapes, containing other flower types which are treated as negative examples for the CNN training. As seen in the detection results (fig. 14), no false positive detections are generated on other types of vegetation.
However, the system can be easily retrained on other flower types of interest, by collecting image material of that flower type. We added some text to clarify this further.

  • Consider different arrangements of sunflowers (simple grid patterns) to simulate sunflowers planted in rows vs grown in clusters in the field.

We agree that this is an interesting field experiment, but that goes far beyond the scope of the paper (which is now more clearly defined, as described above).

  • How does sunflower orientation (sunflowers are level, angled upward or downward) impact the success rate of pollination?

As described above, the pollination success as such is not exactly measured or aimed for. We call our system successful if the drone is able to fly autonomously to the flower and touch the flower’s center. In our detection and angle estimation model, flowers with all vertical orientations are included. As described in section 3.1, we gathered flowers of all of these orientations by rendering 3D models or rotating real or synthetic flowers physically on our chroma key set-up. This enables the detection of these flowers, also if they are angled up or down. We added a sentence to clarify this in the text, and added a photo of this 2 DoF rotator device to figure 6.

  • What statistics can you provide related to timing and the various control measures needed for each successful pollination? (i.e., is the drone struggling and taking a long time to pollinate or is it happening relatively quickly?)

Thank you for this suggestion. We added these numerical results to section 4.3. In order to avoid navigation towards false positive detections, the detection threshold is set a little strict. This means that sometimes in the beginning of a navigation trial, we observe that the drone sometimes is indeed wandering until a stable flower detection is found. This behavior is observed in the (also newly added) demonstration sequence nb 6: Initially, the flower is not detected, causing the drone to rotate around its position until a stable flower detection is present and the actual approach is started.

  • If you have multiple objects and other types of vegetation lined up with the sunflowers, how well does the drone determine the sunflowers vs the other objects? Use multiple trials and statistics to share quantitative results. Need experimental results comparing sunflower detection to other types of flowers/vegetation/objects to determine how well the algorithm has learned sunflowers from other objects.

This would indeed be an interesting experiment. However, we believe that the neural network trained is very good at discriminating different species. In other projects, we have also seen that. In our evaluation image dataset, we see that the detection AP is for several models around 0.70 (see the newly added table 1), which indicates that the number of false positive and false negative detections is quite small. In the lab experiments, false detections never occured. Unfortunately, we do not have time for this lab-scale paper to extend it towards full field tests for the major revision. We keep that for future work.

  • What happens when the wind from the drone blows the flower or when winds move the flower? How do your algorithms handle this volatility? This needs to be addressed in the manuscript.

Again, we think this is interesting to know, but investigation of this falls outside of the scope of this paper. The resistance to wind blows is more related to the stability of the drone’s autopilot, and not the visual servoing control loop above it. We selected the DJI Robomaster TT especially for its decent stability in its size class. It has a stability ensuring IMU based on both inertial sensors as well as a ground feature tracker. We clarified the scope of the paper better, such that the reader knows what experimental validation to expect.

  • Please explain mAP of 0.36 for the reader in the manuscript.

Thank you for this suggestion. We clarified both AP and mAP in the text now, see section 4.1.

  • The videos show the drone approaching the sunflower but the head of the sunflower is not touched in either case. Is this considered a successful pollination? In reality, a successful pollination occurs when a bee lands on the head of the sunflower and drags pollen from its legs onto the stigma in the head of the plant. Just touching the petals (ray florets) will not induce pollination. This needs to be clarified throughout. 

We thank the reviewer for pointing that out. This effect was caused by the fact that our direct steering visual navigation model tries to align the flower’s center with the camera. As the camera is mounted high on our prototype, this caused a vertical position error for the pollination rod. For the major revision of this paper, we solved this error by implementing a vertical bias correction. In the newly conducted experiments (sequences 3 to 6 in table 2, also visible in figure 16), this position difference is taken into account, resulting in more correct “pollinations”, where the rod exactly touches the center of the flower.

  • Also, could your training dataset be improved so that your detection algorithms focus on the sunflower head instead of the sunflower in general? This will more accurately replicate successful pollination.

Indeed, it is certainly important to touch the center of the flower. We believe that, with the bias correction described above, this is the fact now. If for another flower species, for instance, another part of the flower must be touched, we can easily change that. The direct steering model is trained with close-up images of the flower, which can be chosen to have any part as center position.

  • This conclusion is misleading based on the brevity and incompleteness of Section 4. I anticipate the conclusions and discussion will be much improved once satisfactory experimental design and results are implemented.

We totally agree with this remark. Both the described goal of the paper (fully successful pollinations) and the meager amount of experiments conducted towards that goal were not in-line. We believe with the clarification of the goal of the paper, and the severe extension of the experimental section (what was possible in the allotted time frame), we hope this is more satisfactory in this revised version.

Round 2

Reviewer 1 Report

The manuscript has been adjusted and supplemented according to the author's comments.

Some of my questions have been solved through Section 3.3.3, Section 4.1, Figure 14, Figure 16 and Table 2 supplemented by the author. In addition, there are some questions and suggestions:

  1. If there are multiple flowers (flower 1, flower 2, flower 3, ...) waiting for pollination, how to find flower 2 after pollination of flower 1 is finished?
  2. The decrease in quantization bits (32-bit floats to 8-bit ints) may lead to a decrease in the model detection AP. Can you show the comparison results of the optimized model and the pre-optimized model in Section 3.3.3? It doesn't matter if it's lower.

Thank you very much.

Author Response

> If there are multiple flowers (flower 1, flower 2, flower 3, ...) waiting for pollination, how to find flower 2 after pollination of flower 1 is finished?
Dear reviewer, thank you for this question. Indeed, this aspect is left quite vague in the manuscript, as we did not do experiments yet on navigation cases with multiple flowers. However, after pollinating one first flower, we plan to let the drone do a random maneuver (consisting of a backward and upward movement, random horizontal rotation and an adjustable horizontal movement). Then, the drone will be in a new random position from where it can start approaching the closest sunflower. This random motion will eventually make it possible to visit multiple flowers in a field.
The alternative, building a 3D map of flowers and performing SLAM within that map is left for future work.
We added these clarifications to the paper (see section 3.3.2).

> The decrease in quantization bits (32-bit floats to 8-bit ints) may lead to a decrease in the model detection AP. Can you show the comparison results of the optimized model and the pre-optimized model in Section 3.3.3? It doesn't matter if it's lower.
That is certainly correct. For the sake of completeness, we added experimental results on the achieved accuries before and after quantization for several neural network architectures of our direct steering model (see section 4.2). Unfortunatly, we did not have time to add similar results for the flower detection models, but the drop in accuracy will have a comparable magnitude there.